# Review and a Theoretical Approach on Pressure Drop Correlations of Flow through Open-Cell Metal Foam

**DOI:** 10.3390/ma14123153

**Published:** 2021-06-08

**Authors:** Huizhu Yang, Yongyao Li, Binjian Ma, Yonggang Zhu

**Affiliations:** Center for Microflows and Nanoflows, School of Mechanical Engineering and Automation, Harbin Institute of Technology, Shenzhen 518055, China; yy-li@stu.hit.edu.cn (Y.L.); mabinjian@hit.edu.cn (B.M.)

**Keywords:** pressure drop, metal foams, permeability, Forchheimer coefficient, mathematical models

## Abstract

Due to their high porosity, high stiffness, light weight, large surface area-to-volume ratio, and excellent thermal properties, open-cell metal foams have been applied in a wide range of sectors and industries, including the energy, transportation, aviation, biomedical, and defense industries. Understanding the flow characteristics and pressure drop of the fluid flow in open-cell metal foams is critical for applying such materials in these scenarios. However, the state-of-the-art pressure drop correlations for open-cell foams show large deviations from experimental data. In this paper, the fundamental governing equations of fluid flow through open-cell metal foams and the determination of different foam geometry structures are first presented. A variety of published models for predicting the pressure drop through open-cell metal foams are then summarized and validated against experimental data. Finally, two empirical correlations of permeability are developed and recommended based on the model of Calmidi. Moreover, Calmidi’s model is proposed to calculate the Forchheimer coefficient. These three equations together allow calculating the pressure drop through open-cell metal foam as a function of porosity and pore diameter (or strut diameter) in a wide range of porosities *ε* = 85.7–97.8% and pore densities of 10–100 PPI. The findings of this study greatly advance our understanding of the flow characteristics through open-cell metal foam and provide important guidance for the design of open-cell metal foam materials for different engineering applications.

## 1. Introduction

Metal foam is an extremely light-weight porous material consisting of a mesh-like solid metal matrix with randomly distributed pores/voids. Depending on the connection condition between the voids, the metal foam can be generally categorized as either closed-cell or open-cell structure. In closed cell foams, the voids are sealed and separated by solid walls (Figure 1a). In the open-cell foams, the voids are connected via an interconnected network which provide tortuous flow passages with irregular shapes (Figure 1b). The structure of open-cell metal foams can be represented by a collection of dodecahedron-like cells with 12–14 pentagonal or hexagonal faces. The solid ribs around the pores are known as struts (or ligaments), which interconnect the neighboring nodes. The specifications of open-cell metal foam are often provided in terms of two parameters by most manufacturers: (1) the volumetric porosity, *ε*, which is defined as the ratio of the volume of the void space to the total volume of the porous material; and (2) the pore density, which is measured in pores per inch (PPI). Commercially available open-cell metal foams are often made of aluminum, nickel, copper, titanium, iron, steel, magnesium, lead, and their alloys. Open-cell metal foams typically retain the intrinsic physical properties of their base material. In addition to this, many unique characteristics have also been found, including [1,2,3,4]:

A high porosity (*ε* ≥ = 0.85–0.99);A low density (e.g., 0.45 kg/m^3^) and light weight;A high interstitial surface area per unit volume, which ranges from 1000 to 3000 m^2^/m^3^. Such a value can reach up to 8000 m^2^/m^3^ under compressed condition;Excellent fluid mixing, due to the tortuous flow paths;A high permeability.

With these distinctive properties, metal foams have been widely used in a diverse range of industrial applications, such as many heat transfer systems [6,7,8,9,10,11,12], solar energy collection systems [13,14,15], and chemical and nuclear engineering [16]. A fundamental aspect for these applications is the flow characteristics through the open-cell metal foams, which directly determine the heat and mass transfer performance, chemical reaction rates, and filtration efficiency. The topic of fluid flow in porous media has been studied extensively, with a plethora of literature over past few decades [17,18]. However, the majority of the literature has focused on traditional and natural porous media, with the porosity *ε* in a range of 0.4–0.6, such as packed beds, rocks, sand, limestone, brick, wood, coal, cork, fiberglass, soil, and bone.

The last two decades have witnessed a sharp rise in the number of engineering applications and research studies associated with open-cell metal foams. In addition to the development of novel manufacturing techniques, the transport behavior of fluids through the porous metal foams has also been explored by a considerable number of works [19,20,21,22,23,24]. However, a large discrepancy has been found between different studies and different theoretical models on the flow behavior, such as the correlation between flow rate and pressure drop. The inconsistent research findings on fluid flow through open-cell metal foams significantly jeopardize their application.

Therefore, this paper aimed to develop new pressure drop correlations for open-cell metal foam that provide an enhanced applicability and adaptability. The first section presents the fundamental governing equations of fluid flow through open-cell metal foams and the determination of different foam geometry structures. The second section provides a review on the empirical and analytical models of the pressure drop through open-cell metal foams based on the permeability, the Forchheimer coefficient, and the friction factor. In the third section, several permeability and Forchheimer coefficient models proposed in past studies are validated against the experimental data collected in the open literature. Based on these findings, revisions of some of the models are proposed to improve the accuracy. Finally, three correlations for permeability and the Forchheimer coefficient are developed and recommended for calculating the pressure drop through an open-cell metal foam.

## 2. Governing Equations of Fluid Flow and Geometry Structures of Open-Cell Metal Foams

### 2.1. Governing Equations

Volume averaging of the conservation of mass for a fluid flowing in a porous medium leads to the following:(1)$$\varepsilon \frac{\partial \rho }{\partial t}+\nabla \left(\rho \nu \right)=0$$
where *ε* is the porosity, *ρ* is the density of the fluid, *t* is time, and *ν* is the velocity vector. The difference between Equation (1) and the common equation of continuity is the presence of the porosity *ε* and the fact that the velocity vector *ν* is volume-averaged [25].

For laminar flow of an incompressible Newtonian fluid in a porous medium, the volume averaged momentum equations for forced convection are expressed as [26]:(2)$$\rho \left[\frac{1}{\varepsilon }\frac{\partial \nu }{\partial t}+\frac{1}{\varepsilon^{2}}\left(\nu \nabla \nu \right)\right]=-\nabla p+\frac{\mu }{\varepsilon }\nabla^{2}\nu -\frac{\mu }{K}\nu -\frac{c\rho }{\sqrt{K}}u\nu$$
where *u* is the magnitude of the velocity vector, i.e., *u* = |*ν*|, *p* is the static pressure, *μ* is the dynamic viscosity, *K* is the permeability with a unit of length square (m^2^), and *c* is the Forchheimer coefficient. The second term on the right-hand side (RHS) accounts for the viscous shear stress inside the fluid, including the shear stress next to any solid confining walls, if present [27]. The third term on the RHS accounts for viscous dissipation due to flow over the surfaces of the solid structure of the porous medium, while the fourth term (usually called the Forchheimer term) accounts for the form drag effect.

A creeping (very slow) flow of a fluid through a porous medium in steady-state condition can be described by Darcy’s law, where the pressure-drop is proportional to the product of the fluid velocity and the dynamic viscosity, and inversely proportional to the permeability, i.e.,
(3)$$\frac{\Delta p}{L}=\frac{\mu }{K}\nu$$

For high velocity flows in a steady-state condition, the relationship between the pressure drop and the flow rate is nonlinear and the form drag term becomes significant. In this case, the pressure drop through the porous medium is given by the Forchheimer equation as:(4)$$\frac{\Delta p}{L}=\frac{\mu }{K}\nu +\rho Cu\nu =\frac{\mu }{K}\nu +\rho \frac{c}{\sqrt{K}}u\nu$$
where *C* is a form drag coefficient having units of one over length (m^−1^).

The velocity term *u* = |*ν*| in the above equations can be taken as the Darcy velocity of the fluid flow
(5)$$u_{D}=\frac{Q}{A_{cor}}$$
where *Q* is the volume flow rate and *A*_cor_ is the cross-section area of the channel. The velocity *u* in Equation (4) can also be taken as the pore (filter) velocity given by the Dupuit–Forchheimer relation [28]
(6)$$u_{p}=\frac{u_{D}}{\varepsilon }$$

This relation accounts for the presence of the solid phase in the channel by dividing the Darcy velocity over the volumetric void fraction of the medium (assuming an isotropic medium).

Either velocity can be used for characterizing and deriving the permeability and Forchheimer coefficient [29,30].

Both the Darcy and Forchheimer models are derived for laminar flow. Turbulent flow in porous media is highly chaotic. According to the studies of Fand et al. [31], Kececioglu and Jiang [32], Seguin et al. [33], Skjetne and Auriault [34], and N. Dukhan et al. [35,36], the pressure drop characteristic of the porous media in turbulent flow can be still described by the Forchheimer equation, with the introduction of different fitting parameters (different permeability and Forchheimer coefficient). Meanwhile, for fully developed turbulent flow at sufficiently high flow velocity, the effect of viscous drag becomes negligibly, and the form drag term becomes dominant.

The common method of determining permeability *K* and Forchheimer coefficient *c* (or form drag coefficient *C*) is by curve-fitting measured pressure gradients using Equations (3) and (4).

As mentioned earlier, the volume-averaged momentum equation is valid for an incompressible Newtonian fluid. However, for gas flows, the static pressure variations over the length of the metal foam can be significant enough to alter the density of the flowing gas. Therefore, the effect of compressibility should be taken into account when determining the permeability and the Forchheimer coefficient by Equations (3) and (4). Otherwise, the pressure drop will be underestimated significantly [37]. Considering that Equation (4) still holds for a gas flow over a small distance along the flow direction, the compressibility effect can be taken into consideration by integrating Equation (4) from the entrance (*x* = 0, *p* = *p*_o_) to the exit (*x* = *L*, *p* = *p*_L_) of the metal foam, i.e.,
(7)$$\int_{p_{o}}^{p_{L}}-dp=\int_{0}^{L}\left(\frac{\mu }{K}u+\frac{\rho c}{\sqrt{K}}u^{2}\right)dx$$

Based on the ideal gas law $\frac{p}{\rho }=\frac{RT}{M}$ and mass conservation *G* = *ρu*, Equation (7) can be simplified as:(8)$$\int_{p_{o}}^{p_{L}}-pdp=\int_{0}^{L}\left(\frac{\mu }{K}Ge+\frac{c}{\sqrt{K}}G^{2}e\right)dx$$
(9)$$e=\frac{RT}{M}$$
and:(10)$$\frac{p_{o}^{2}-p_{L}^{2}}{2Le}=\frac{\mu }{K}G+\frac{c}{\sqrt{K}}G^{2}$$
where *R* is the universal gas constant, *T* is absolute temperature, *M* is molecular weight, and *G* is the mass flow rate. Thus, when analyzing the flow of gas through open-cell metal foams, the permeability *K* and Forchheimer coefficient *c* should be determined by plotting $\frac{p_{o}^{2}-p_{L}^{2}}{2Le}$ against *G* [38].

### 2.2. Geometry Structure

An accurate description of the foam geometric structure is a prerequisite for assessing the fluid flow transport through the open-cell metal foams. The geometric parameters of the foams can be obtained by two methods: direct measurement and building an ideal geometric model. The 3D structure of open-cell metal foam can be directly measured by micro computed tomography (*μ*CT) scan and scanning electron microscope (SEM) [39,40,41,42]. During *µ*CT-scan, the solid structure is sliced virtually into parallel slabs with a constant interval. A complete 3D model of the foam’s structure can then be reconstructed after stacking the digital slices. Figure 2 depicts the 3D structure of two in-house manufactured foams from the *µ*CT-scan and reconstruction. Scanning electron microscope (SEM) can be used to measure the pore and strut diameters of metal foam. Richardson et al. [39] detected the individual pore areas with Jandel SigmaScan software in the SEM image of open-cell metal foams, and used the Gaussian normal and log-normal functions to fit the pore diameter distribution, from which the average pore and strut diameters were determined. Both *μ*CT-scan and SEM can provide an accurate geometric structure of the open-cell metal foam, at a high cost.

Alternatively, one may approximate the structure of an open-cell metal foam by foam cell geometry idealization. In most studies, the structure of a unit cell in the open-cell foam is modeled by polyhedrons [43,44,45,46]. Based on the shape of a polyhedron, the geometric models of open-cell metal foam can be divided into four types: the cubic unit cell model, dodecahedron model, tetrakaidecahedron model, and body-centered-cubic model. In the cubic unit cell model, a unit cell is described by three perpendicular struts located at the cell’s center. The dodecahedron model consists of twelve pentagonal faces. The tetrakaidecahedron model consists of six square faces and eight hexagonal faces. In the body-centered-cubic model, the geometric shape of the foam is obtained by subtracting the unit cell cube from the spheres sharing the same centers and vertices. However, the radius of the sphere must be larger than half of the length of the cube in the open-cell metal foam. Therefore, the body-centered-cubic model is only applicable for open-cell metal foams with a porosity greater than 94%. Table 1 summarizes the typical geometric parameters (pore/struts diameter, ligament length, porosity, and specific surface area) of open-cell metal foams for the four different models.

## 3. Pressure Drop Correlations for Open-Cell Metal Foams

Ergun (1952) proposed one of the first empirical equations for calculating the pressure drop in porous media [47] as:(11)$$\frac{\Delta p}{L}=A\frac{{\left(1-\varepsilon \right)}^{2}\mu }{\varepsilon^{3}d_{par}^{2}}u+B\frac{\left(1-\varepsilon \right)\rho }{\varepsilon^{3}d_{par}}u^{2}$$
where *A* and *B* are empirical constants and *d*_par_ is the particle diameter. This equation was originally established for porous media consisting of packed spheres where *A* and *B* are set as 150 and 1.75, respectively. However, open-cell metal foams exhibit a very high porosity and a web-like internal structure, with distinctively different features from the packed spheres. Therefore, existing correlations developed for packed beds and granular porous media cannot be applied directly to metal foams. The pressure drop of fluid flow through open-cell metal foams has only been explored by a limited number of studies. Those studies were mostly based on two methods. The first method uses the Darcy–Forchheimer equation (i.e., Equation (4)) with permeability *K* and Forchheimer coefficient *c* to estimate the pressure drop, while the second method considers the friction factor in the foam to estimate the pressure drop, which is usually expressed as a function of the Reynolds number Re. A detailed review of these studies is provided in the following section. A collection of different pressure drop correlations for open-cell metal foam are summarized in Table 2 and Table 3.

### 3.1. Permeability and Forchheimer Coefficient

The correlations of permeability and Forchheimer coefficient are summarized in Table 2. Dukhan [48] proposed correlations to calculate permeability and form drag coefficient as functions of the porosity *ε*. The empirical constants *c*_1_, *c*_2_, *c*_3_, and *c*_4_ were obtained by fitting the experimental data obtained with air at a Darcy velocity in the range of 0.75 to 3.75 m/s and a foam porosity in the range of 68.2% to 92.4%. Richardson et al. [39] developed permeability and form drag coefficient correlations based on Ergun’s model. In their correlations, the reciprocal of the surface area per unit volume *S_v_* is taken as the characteristic length. In addition, the parameters *A* and *B* are set to vary with different porosities *ε* and pore diameters *d_p_*. Similarly, Tadrist et al. [49] developed new correlations of permeability and form drag coefficient based on the Ergun equation, in which the strut diameter *d_f_* is taken as the characteristic length. The coefficients *A* vary between 100 and 865, while the coefficient *B* varies between 0.65 and 2.6. Dukhan et al. [50] proposed the Ergun-like relation by using the average strut diameter *d_f_* as the characteristic length. The effect of other geometrical parameters of the foam was supposedly captured in the exponents *m* and *n*. The empirical constants *A*, *B*, *m,* and *n* were obtained from experiments of the air flow through compressed and uncompressed open-cell aluminum foam.

Using the cubic unit cell model, Du Plessis et al. [51] established a theoretical model for predicting the permeability and form drag coefficient as functions of porosity *ε*, the width of the cubic *d*, and tortuosity χ. Similarly, Fourie and Du Plessis [43] developed a correlation for Forchheimer coefficient as a function of porosity *ε*, tortuosity χ, and coefficient *C_D,F_*. Bhattacharya et al. [52] established a correlation for Forchheimer coefficient as a function of porosity *ε*, tortuosity χ, geometric function *G*, and coefficient *C*_D_. The permeability model proposed by Du Plessis et al. [51] was used in both these correlations. Calmidi [19] proposed correlations for permeability and Forchheimer coefficient as functions of the porosity *ε* and the ratio between the strut diameter *d_f_* and pore diameter *d_p_*. The parameter *d_f_*/*d_p_* characterizes the strength of drag with different strut diameters and pore sizes. Recently, Yang et al. [53] proposed an analytical equation for permeability as a function of the porosity *ε* and the width of the cubic *d,* based on the cubic unit cell model. Their equation can estimate the permeability over a wide range of porosities *ε* = 55–98% and pore densities between 5–100 PPI.

### 3.2. Friction Factor

In general, the friction factor *f* and Reynolds number Re are defined as:(12)$$f=\frac{\Delta p\ell_{f}}{L\rho u^{2}}$$
and
(13)Re=ρuℓfμ
where $\ell_{f}$ is the characteristic length. The relationships between the friction factor and the Reynolds number are summarized in Table 3. By taking the square root of permeability as the characteristic length, Kamath et al. [54] developed an empirical correlation between friction factor *f_K_* and Reynolds number, in which the coefficient *c_F_* equals 0.129 for aluminum foam and 0.147 for copper foam. Similarly empirical correlations have been developed by Paek et al. [55], Vafai and Tien [26], Noh et al. [56], Beavers and Sparrow [57], and Hamaguchi et al. [58], where the coefficient *c_F_* was set as 0.105, 0.057, 0.05, 0.074, and 0.076, respectively. The correlations proposed by Paek et al., Vafai and Tien, and Noh et al. were developed for aluminum-based metal foams whiles those by Beavers and Sparrow and Hamaguchi et al. were developed for nickel-based metal foams. The structure and ligament thickness of metal foams vary with different materials, due to their distinct manufacturing processes. Such variation leads to different values of *c_F_*. However, if we rewrite Equation (4) as a correlation between the friction factor and the Reynolds number, it will be found that *c_F_* is nothing but the Forchheimer coefficient, which should be strongly dependent on the foam structure. Therefore, it is more rational to have the coefficient *c_F_* as a function of the foam structure rather than a constant. Hwang et al. [59] presented a friction factor correlation as a function of the Reynolds number. The length scale of friction factor and Reynolds number are one and the length of metal foam, respectively. The values of *a* and *b* vary with different porosities and are determined by experimental testing of air flow through 10 PPI aluminum foam. Based on the Ergun model, Dukhan and Petal [60] developed a friction factor correlation as a function of the Reynolds number and porosity, in which the reciprocal of the surface area density *S_v_* is taken as the characteristic length. The constants *A* and *B* used in their equation vary with different metal foams. Liu et al. [27] proposed a friction factor correlation based on the pressure drop measurement through aluminum and ceramic foams, with the porosity ranging from 0.802 to 0.958 and pore density between 5 and 65 PPI.

## 4. Discussions

In this section, the existing permeability and Forchheimer coefficient correlations are validated against the experimental data collected in the open literature [7,27,35,43,48,52,54,61,62,63,64,65,66,67]. The experimental data used for the validation were measured for air and water flow through aluminum and copper foams. Based on the comparison between the correlations and experimental measurements with the nonlinear curve fit method, improvements of existing empirical correlations are proposed to enhance their estimation accuracy. The revised correlations for permeability and Forchheimer coefficient provide a better fit with the experimental values.

The performance of each permeability and Forchheimer coefficient model was assessed based on the Root of Mean Squared error (RMS) defined as:(14)$$\mathrm{RMS}=\sqrt{\frac{1}{n}{\sum_{i=1}^{n}\left(y_{i}-y_{i,\exp}\right)}^{2}}$$
where *y_i_* and *y_i_*_,exp_ are the predicted and measured values, respectively, and *n* is the total number of data points.

Figure 3 shows the change in the ratio of the permeability over the square of the average pore diameter (*K*/*d_p_*^2^) with increasing porosity from selected experimental data and permeability models, constructed as functions of porosity and mean pore diameter *d_p_*. As shown in Figure 3a, the experimental data from Liu et al. [27] are significantly higher than the rest of the experimental data. The ratio of the permeability over the square of the average pore diameter (*K*/*d_p_*^2^) is substantially overestimated by Du Plessis’s model [51]. In addition, Du Plessis’s model shows a decreasing trend of *K*/*d_p_*^2^ with porosity. However, it is understandable that the flow cross-section area will increase, with the porosity increasing, when maintaining the pore diameter, and hence the flow resistance will be reduced and a higher permeability value will be obtained. Therefore, the value of *K*/*d_p_*^2^ should increase with increasing porosity, as shown by the models of Richardson et al. [39], Calmidi [19], and Yang et al. [53]. Figure 3b shows the data comparison, after excluding the experimental data of Liu et al., and the trend predicted by Du Plesssis’s model. It is found that the value of *K*/*d*^2^ is overestimated by Yang’s model, while it is underestimated by Richardson’s model. Furthermore, the prediction from Richardson’s model shows a high level of fluctuation in *K*/*d*^2^ with increasing porosity. Such a behavior is attributed to the fact that the coefficient *A* used in Richardson’s model is defined as a function of porosity and mean pore diameter. The model proposed by Calmidi agrees very well with the experimental data with a RMS of 0.0051. However, it is interesting to note that when the porosity is larger than 0.96, the value of *K*/*d_p_*^2^ decreases with the increasing porosity (shown as by the dotted circle in Figure 3b). Such a feature is contradictory to the actual physics.

To improve the accuracy of empirical correlation, the empirical constants in the expression of coefficient *A* in Richardson’s model and the empirical constants in the right-hand side of Calmidi’s equation are revised using the least square methods. The modified equations are given by:(15)$$A=12.757d_{p}^{0.126}{\left(1-\varepsilon \right)}^{0.149}$$
and:(16)$$\frac{K}{d_{p}^{2}}=0.00747{\left(1-\varepsilon \right)}^{-0.2329}{\left(d_{f}/d_{p}\right)}^{0.00249}$$

Figure 4 shows a comparison of Richardson’s model and Calmidi’s model, before and after revising the empirical constants. As shown in the figure, the improved Richardson’s model and improved Calmidi’s model provide highly consistent results with the experimental measurements, where the RMS for each model are 0.0048 and 0.00049, respectively. Furthermore, the trend predicted by the improved Calmidi’s model shows a monotonically increasing permeability with the porosity.

Figure 5 shows the comparison between the experimental data and the permeability models constructed as a function of porosity and mean strut diameter. The models of Dukhan et al. [48] and Tadrist et al. [49] fail to capture the trend of the experimental measurements accurately. The RMS of Tadrist’s model is 0.816 even with an optimized empirical constant of *A* = 645. Meanwhile, the RMS of Dukhan’s model is 1.034.

To provide a better prediction accuracy, the empirical coefficients used in Dukhan’ model and Tadrist’s model are modified using the least square method. In particular, the coefficient *A* used in Tadrist’s model is expressed as a function of mean strut diameter and porosity, a similar approach as was taken to modify Richardson’s model. In addition to improving the existing models, a new permeability model has also been proposed in our study based on Calmidi’s model. In our proposed model, the permeability is expressed as a function of the strut diameter *d_f_*, parameter *d_f_*/*d_p_*, and porosity *ε*. The modified equations and our proposed model are given by:(17)$$K=\frac{1}{8179145.4}{\left[\frac{\varepsilon^{3}d_{f}}{{\left(1-\varepsilon \right)}^{2}}\right]}^{0.13865}$$
(18)$$A=1.182d_{f}^{-0.013}{\left(1-\varepsilon \right)}^{-1.834}$$
and
(19)$$\frac{K}{d_{f}^{2}}=0.3199{\left(1-\varepsilon \right)}^{-0.3577}{\left(d_{f}/d_{p}\right)}^{0.02234}$$

Figure 6 shows a comparison between the experimental data, Dukhan’ model and Tadrist’s model, before and after revising the empirical constants, and the permeability model proposed in this study. The permeability values predicted by the improved Dukhan’s model, the improved Tadrist’s model, and the proposed model agree well with the experimental data. In addition, similar fluctuation behavior is observed in the permeability curve from the improved Tadrist’s model due to the fact that the coefficient *A* is a function of porosity and strut diameter. Compared with the experimental data, the RMS of the improved Dukhan model, improved Tadrist model, and the proposed permeability model were 0.656, 0.405, and 0.411, respectively.

A comparison of the Forchheimer coefficients between the experimental data and the empirical models is plotted in Figure 7. The Forchheimer coefficient was overestimated substantially by the model of Dukhan et al. [50]. However, the rest of the models from Du Plessis et al. [51], Bhattacharya et al. [52], and Calmidi [19] provide pretty consistent predictions with the experimental data. The RMS of the models from Dukhan et al., Du Plessis et al., Bhattacharya et al., and Calmidi were 0.189, 0.062, 0.08, and 0.052, respectively. Therefore, Calmidi’s model is recommended as the best model for estimating the Forchheimer coefficient of open-cell metal foams.

In order to examine the prediction ability of the above-mentioned permeability correlations, the predicted values were compared with the experimental data in Figure 8. As mentioned above, the permeability was overestimated substantially by the correlation of Yang et al. [53] and Du Plessis [51], while it was underestimated substantially by the correlations of Richardson et al. [39], Tadrist et al. [49], and Dukhan et al. [50]. However, Calmidi’s correlation and the improved correlations provided relatively consistent results with the experimental measurements, where most of the relative errors were less than 100%. The RMS values of these correlations are presented in Table 4. The RMS values for the improved correlations presented relatively small values. Therefore, the comparison results indicated that the improved correlations provided more reliable prediction in a wider range, and the improved correlations based on Calmidi’s model provided the best predictive ability.

According to the above-mentioned analysis, it can be concluded that following the same format as Calmidi’s model of permeability, the following two empirical correlations of permeability were developed and are recommended for open-cell metal foams:(20)$$\frac{K}{d_{p}^{2}}=0.00747{\left(1-\varepsilon \right)}^{-0.2329}{\left(d_{f}/d_{p}\right)}^{0.00249}$$
and:(21)$$\frac{K}{d_{f}^{2}}=0.3199{\left(1-\varepsilon \right)}^{-0.3577}{\left(d_{f}/d_{p}\right)}^{0.02234}$$
where $d_{f}/d_{p}=2\sqrt{\frac{1-\varepsilon }{3\pi }}\frac{1}{G}$ and $G=1-e^{-\left(1-\varepsilon \right)/0.04}$.

Moreover, Calmidi’s model is recommended for calculating the Forchheimer coefficient in an open-cell metal foam:(22)$$c=0.00212{\left(1-\varepsilon \right)}^{-0.132}{\left(d_{f}/d_{p}\right)}^{-1.63}$$

Equations (20)–(22) provide a complete set of correlations to predict the pressure drop through open-cell metal foam based on the porosity *ε* and pore diameter *d_p_* (or strut diameter *d_f_*). These equations are valid for open-cell metal foams with a porosity *ε* ranging from 85.7 to 97.8% and a pore density ranging from 10 to 100 PPI.

## 5. Summary and Conclusions

This paper studied the fluid flow characteristics in open-cell metal foam. The fundamental governing equations of fluid flow through open-cell metal foams and the geometry structure of metal foams were summarized. The different pressure drop models for flow transport through open-cell metal foam were presented and discussed. Improvements were made on selected models to provide a better prediction accuracy against experimental measurements. The following conclusions are drawn from this review:The pressure drop for turbulent flow through open-cell metal foams can still be described by the Forchheimer equation with different fitting parameters (different permeability and Forchheimer coefficient). Meanwhile, for compressible gas flow, the permeability and Forchheimer coefficient should be obtained using Equation (10).There is a significant deviation in the permeability between the predictions from most empirical models (except for Calmidi’s model) and the experimental data available in the literature. Even for Calmidi’s model, an abnormal change of permeability is found for porosities greater than 0.96. However, the empirical Forchheimer coefficient models from Du Plessis et al. and Calmidi were found to yield consistent results with the experimental data.By imitating Calmidi’s model for permeability, two empirical correlations of permeability were developed in this study and recommended for open-cell metal foam. Meanwhile, Calmidi’s model is recommended for calculating the Forchheimer coefficient for flow through open-cell metal foam. These three equations together allow calculating the pressure drop through open-cell metal foam as a function of porosity and pore diameter (or strut diameter) for a wide range of porosities *ε* between 85.7 and 97.8% and pore densities between 10 and 100 PPI.

## Figures and Tables

**Figure 1 materials-14-03153-f001:**
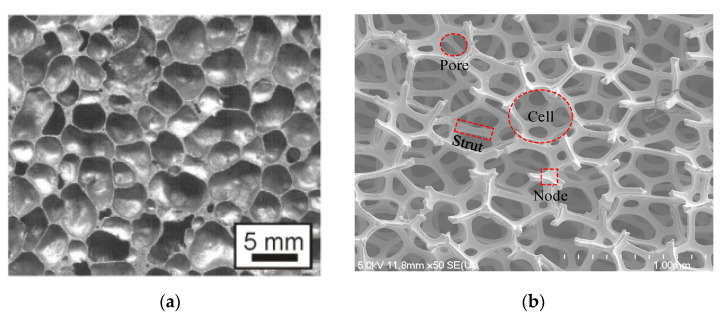
The SEM images of closed-cell and open-cell metal foams: (**a**) closed-cell foam [5]; (**b**): open-cell foam.

**Figure 2 materials-14-03153-f002:**
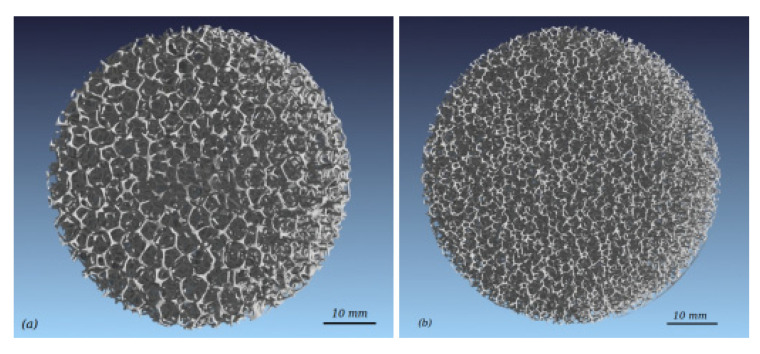
*µ*CT-scan reconstruction of metal foam [40]: (**a**) 10 PPI; (**b**) 20 PPI.

**Figure 3 materials-14-03153-f003:**
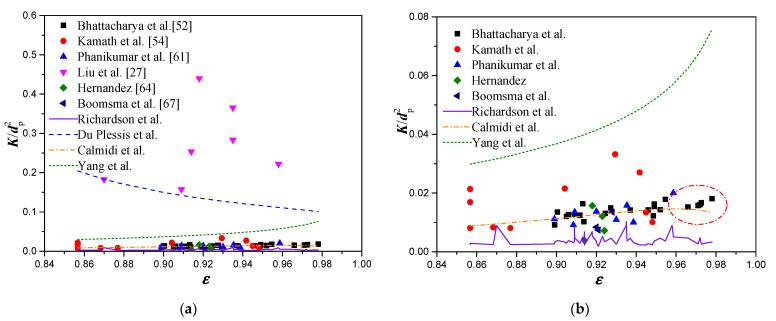
Comparison between experimental data and permeability models constructed as functions of porosity and mean pore diameter: (**a**) all data; (**b**) all data excluding the Liu et al. and Du Plesssis et al.

**Figure 4 materials-14-03153-f004:**
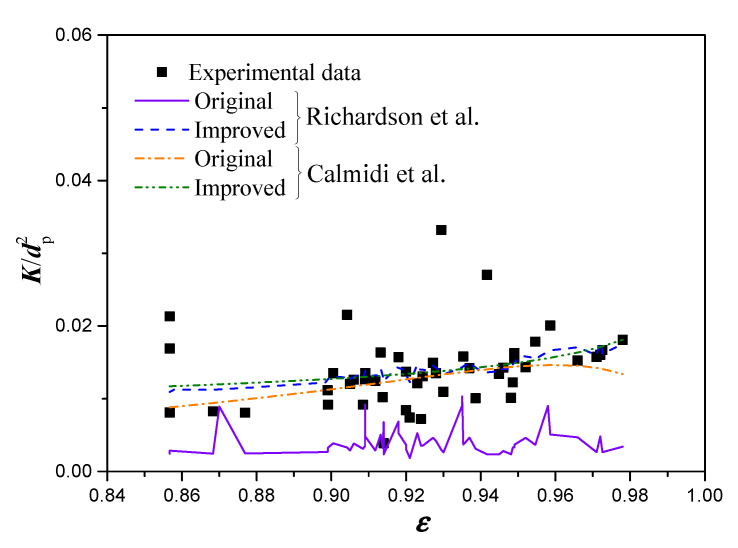
Comparison of the permeabilities predicted by Richardson’s model and Calmidi’ model, before and after revising the empirical constants.

**Figure 5 materials-14-03153-f005:**
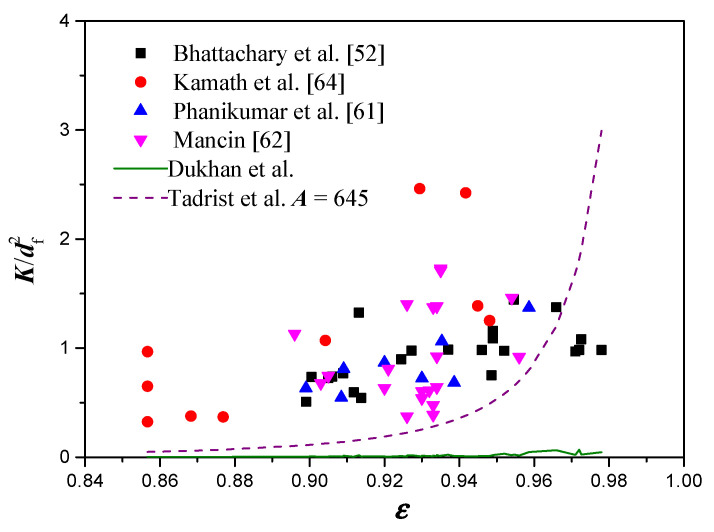
Comparison between experimental data and permeability models constructed as a function of porosity and mean strut diameter.

**Figure 6 materials-14-03153-f006:**
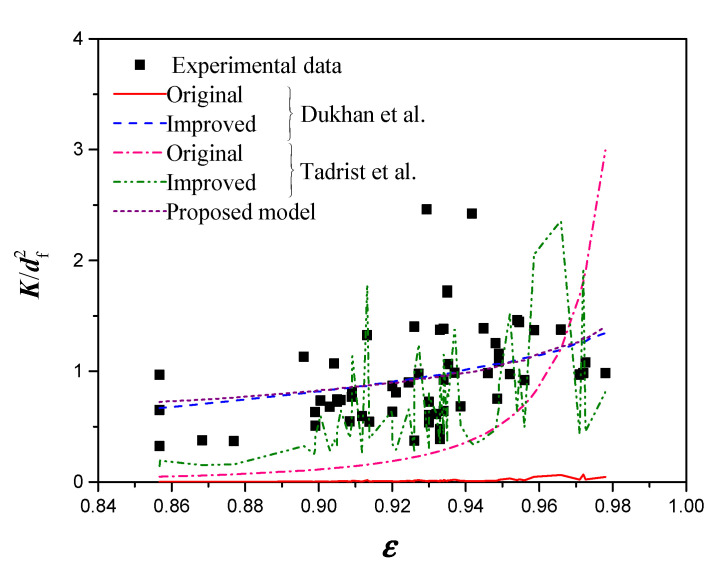
Comparison of the permeability predicted by Dukhan’s model and Tadrist’s model before and after revising the empirical constants, and the proposed model.

**Figure 7 materials-14-03153-f007:**
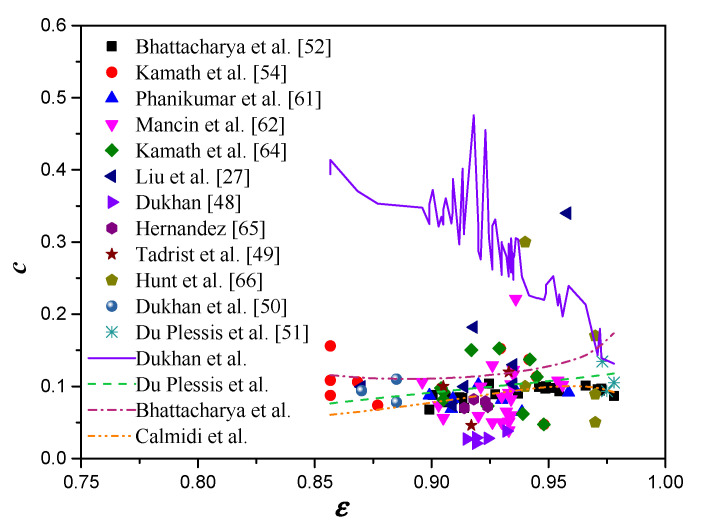
Comparison of Forchheimer coefficients between experimental data and different empirical models.

**Figure 8 materials-14-03153-f008:**
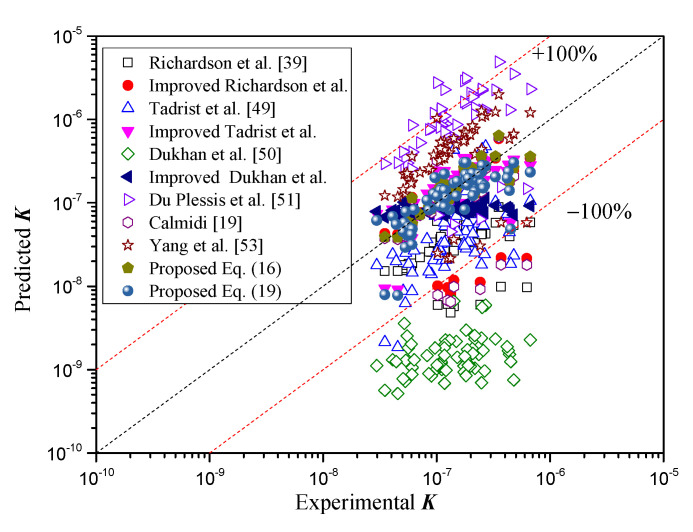
Comparison of permeability between experimental data and the above-mentioned correlations.

**Table 1 materials-14-03153-t001:** Geometric relationships of open-cell metal foams.

References	Unit Cell	Geometric Relationships	Comments
Fourie and Du Plessis [43]	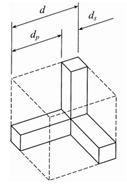 Cubic	$\chi =2+2\cos\left[\frac{4\pi }{3}+\frac{1}{3}\cos^{-1}\left(2\varepsilon -1\right)\right]$, $d_{p}=\frac{3-\chi }{2}d$, $d_{s}+d_{p}=d$ and $a_{c}=\frac{3}{d}\left(3-\chi \right)\left(\chi -1\right)$	Requires the porosity *ε* and pore diameter *d_p_* as the inputs, and tortuosity *χ* is the important variable
Huu et al. [44]	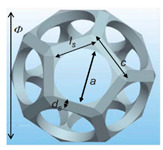 Dodecahedron	$\varphi +1=\varphi^{2}$, $l_{s}=c-\frac{1}{2}\sqrt{\frac{2}{3}d_{s}}$, $\frac{a}{c}=\frac{\varphi }{\sqrt{3-\varphi }}\left(1-\frac{\kappa }{2}\sqrt{\frac{2}{3}}\right)$, $\frac{d_{s}}{\Phi }=\frac{d_{s}}{c\varphi^{2}}$, $\kappa =\frac{d_{s}}{c}$,$\kappa^{2}\frac{\sqrt{15}}{\varphi^{4}}-\kappa^{3}\frac{\sqrt{10}}{3\varphi^{4}}-\left(1-\varepsilon \right)=0$ and $a_{c}=\frac{60k}{\sqrt{5}\varphi^{2}\Phi }\left(1-\frac{1}{2}\sqrt{\frac{2}{3}}\kappa \right)$	For equilateral triangular struts and very high porosity, requires the porosity ε and strut diameter *d_s_* (or pore diameter *a*) as the inputs, and Φ is cell diameter, *c* is side of the perfect pentagon length and *l_s_* is strut length
Kumar and Topin [45]	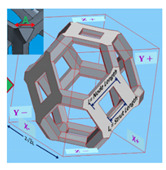 tetrakaidecahedron	$L=0.594A+L_{s}$, $V_{node}=\frac{\sqrt{3}}{3}A^{3}$$V_{ligament}=\frac{\sqrt{3}}{4}A^{2}L_{s}$, $V_{s}=\frac{1}{3}\left(36V_{ligament}+24V_{node}\right)$, $\varepsilon =\frac{1-V_{s}}{V}=\frac{1-\left(3\sqrt{3}A^{2}L_{s}+\frac{8\sqrt{3}}{3}A^{3}\right)}{8\sqrt{2}L^{3}}$ and $a_{c}=\frac{\left(36S_{ligament}+24S_{node}\right)}{V_{c}}=\frac{9\sqrt{2}A}{4L^{4}}\left(L_{s}+\frac{\sqrt{3}}{64}A^{2}\right)$	For equilateral triangular struts and porosity in the range of 80–95%, requires the porosity *ε* and side length of strut *A* (or strut length *L_s_*) as the inputs, and *L* is node length, *V_ligament_* is the volume of the ligament and *V_node_* is the volume of the node
Krishnan et al. [46]	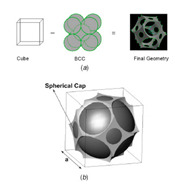 body-centered-cubic	$V_{\mathrm{int}}=\frac{\pi }{12}\left(4R+\frac{\sqrt{3}}{2}a\right){\left(2R-\frac{\sqrt{3}}{2}a\right)}^{2}$, $V_{sc}=\frac{\pi }{3}{\left(R-a/2\right)}^{2}\left(2R+a/2\right)$, $V_{f}=2\left[\frac{4\pi }{3}R^{3}-8\left(\frac{V_{\mathrm{int}}}{2}\right)-6V_{sc}\right]$, $\varepsilon =\frac{V_{f}}{a^{3}}$ and $\begin{array}{l} A_{in,f}=\pi R^{2}+\pi \left(\frac{4R^{2}-a^{2}}{4}\right)\\ -2\left[2R^{2}\cos\left(\frac{a}{4R}\right)-\frac{1}{4}a\sqrt{4R^{2}-{\left(\frac{1}{2}a\right)}^{2}}\right] \end{array}$	Porosity higher than 94%, requires the porosity *ε* and the unit cell length *a* (or radius of sphere *R*) as the inputs, and *A_in,f_* is inlet face area, *V_int_* is the intersection volume between two overlapping spheres and *V_sc_* is the volume occupied by the spherical cap

*a*_c_ is specific surface area.

**Table 2 materials-14-03153-t002:** Permeability and Forchheimer coefficient correlations for open-cell metal foams.

Reference	Expression	Comments
Dukhan [48]	$K=c_{1}\exp\left(c_{2}\varepsilon \right)$, $C=c_{3}\varepsilon +c_{4}$	*c*_1_, *c*_2_, *c*_3_, and *c*_4_ are curve-fit constants that depend on the type of foam.
Richardson et al. [39]	$K=\frac{\varepsilon^{3}}{AS_{v}^{2}{\left(1-\varepsilon \right)}^{2}}$, $C=\frac{BS_{v}\left(1-\varepsilon \right)}{\varepsilon^{3}}$, where $S_{v}=\frac{4\varepsilon }{d_{p}\left(1-\varepsilon \right)}$, $A=9.73\times 10^{2}d_{p}^{0.743}{\left(1-\varepsilon \right)}^{-0.0982}$ and $B=3.68\times 10^{2}d_{p}^{-0.7523}{\left(1-\varepsilon \right)}^{0.07158}$	Requires the porosity ε and pore diameter *d_p_* as the inputs.
Tadrist et al. [49]	$K=\frac{\varepsilon^{3}d_{f}^{2}}{A{\left(1-\varepsilon \right)}^{2}}$, $C=\frac{B\left(1-\varepsilon \right)}{\varepsilon^{2}d_{f}}$	*A* =100–865, *B* = 0.65–2.6. However, it is unclear how the final values of *A* and *B* are selected.
Dukhan et al. [50]	$K=\frac{1}{1.25\times 10^{8}}{\left[\frac{\varepsilon^{3}d_{f}}{{\left(1-\varepsilon \right)}^{2}}\right]}^{0.6155}$, $C=1336.7{\left[\frac{{\left(1-\varepsilon \right)}^{2}}{\varepsilon^{3}d_{f}}\right]}^{0.6184}$	Applicable to uncompressed foam, requires the porosity *ε* and strut diameter *d_f_* as the inputs.
Du Plessis et al. [51]	$\frac{K}{d^{2}}=\frac{\varepsilon^{2}}{36\chi \left(\chi -1\right)}$, $C=\frac{2.05\chi \left(\chi -1\right)}{\varepsilon^{2}d\left(3-\chi \right)}$, where $\frac{1}{\chi }=\frac{\pi }{4\varepsilon }\left[1-{\left(1.18\sqrt{\frac{1-\varepsilon }{3\pi }}\frac{1}{G}\right)}^{2}\right]$, $G=1-e^{-\left(1-\varepsilon \right)/0.04}$ and $d=\sqrt{\frac{\chi }{\varepsilon }}d_{p}$	Developed based on cubic model, requires the porosity *ε* and pore diameter *d_p_* as the inputs.
Fouriea and Du Plessis [43]	$c=\left(3-\chi \right)\left(\chi -1\right)\frac{C_{D,F}\chi^{1.5}}{24\varepsilon^{2}}$ where $C_{D,F}=1+10{\left(\frac{\rho u_{p}d\left(\chi -1\right)}{2\mu }\right)}^{-0.667}$, $\chi =2+2\cos\left[\frac{4\pi }{3}+\frac{1}{3}\cos^{-1}\left(2\varepsilon -1\right)\right]$, $\frac{d_{p}}{d}=\sqrt{\frac{\varepsilon }{\chi }}$	Developed based on cubic model, requires the porosity ε, pore diameter *d*_p_, and pore velocity *u*_p_ as the inputs.
Bhattacharya et al. [52]	$c=0.095\frac{C_{D\left(\varepsilon =0.85\right)}}{12}G^{-0.8}\sqrt{\frac{\varepsilon }{3\left(\chi -1\right)}}{\left(1.18\sqrt{\frac{1-\varepsilon }{3\pi }}\frac{1}{G}\right)}^{-1}$, where $C_{D\left(\varepsilon =0.85\right)}=1.2$, $\frac{1}{\chi }=\frac{\pi }{4\varepsilon }\left[1-{\left(1.18\sqrt{\frac{1-\varepsilon }{3\pi }}\frac{1}{G}\right)}^{2}\right]$, $G=1-e^{-\left(1-\varepsilon \right)/0.04}$ (0.85 < *ε* < 0.97) and *G*=0.5831 (*ε* ≥ 0.97)	Only requires the porosity *ε* as the input.
Calmidi [19]	$\frac{K}{d_{p}^{2}}=0.00073{\left(1-\varepsilon \right)}^{-0.224}{\left(d_{f}/d_{p}\right)}^{-1.11}$, $c=0.00212{\left(1-\varepsilon \right)}^{-0.132}{\left(d_{f}/d_{p}\right)}^{-1.63}$, where $d_{f}/d_{p}=1.18\sqrt{\frac{1-\varepsilon }{3\pi }}\frac{1}{G}$ and $G=1-e^{-\left(1-\varepsilon \right)/0.04}$	Requires the porosity ε and pore diameter *d_p_* (or strut diameter *d_f_* ) as the inputs.
Yang et al. [53]	$\frac{K}{d^{2}}=\frac{\varepsilon {\left[1-{\left(1-\varepsilon \right)}^{1/3}\right]}^{2}}{36\left[{\left(1-\varepsilon \right)}^{1/3}-\left(1-\varepsilon \right)\right]}$, where $d=\sqrt{\frac{\chi }{\varepsilon }}d_{p}$, $\chi =\frac{\varepsilon }{1-{\left(1-\varepsilon \right)}^{1/3}}$	Requires the porosity *ε* and pore diameter *d_p_* as the inputs, applicable for porosity *ε* = 55–98% and pore density between 5–100 PPI.

**Table 3 materials-14-03153-t003:** Friction factor correlations for open-cell metal foams.

Reference	Expression	Comments
Kamath et al. [54]	f=1ReK+cF, where Rek=ρuDKμ, $f=\frac{\Delta p\sqrt{K}}{L\rho u_{D}{}^{2}}$	*c*_F_ = 0.129 for Al; *c*_F_ = 0.147 for Cu
Hwang et al. [59]	f=aReLb, where ReL=ρuDLμ, $f=\frac{2\Delta p}{\rho u^{2}}$	Only valid for 10 PPI, *a* and *b* vary for different porosities
Dukhan and Petal [60]	f=A1−εReσ+B where $f=\left(\frac{\varepsilon^{3}}{1-\varepsilon }\right)\frac{\Delta p\left(1/\sigma \right)}{L\rho u_{D}{}^{2}}$, Reσ=ρuD(1/σ)μ	Requires the surface area per unit volume as the input, *A* and *B* vary with different foam
Liu et al. [27]	Re<30, f∝1−εRe, 30<Re<300, f=221−εRe+0.22 Re>300, f=0.22, where $f=\frac{\Delta pD_{p}}{L\rho u_{D}^{2}}\frac{\varepsilon^{3}}{1-\varepsilon }$, Re=ρuDDpμ(1−ε), $D_{p}=6/S_{v}$, and $S_{v}=\frac{4\varepsilon }{d_{p}\left(1-\varepsilon \right)}$	Requires the porosity ε and pore diameter *d_p_* as the input, valid for porosity ε ranging from 80.2–95.8% and pore density between 5–65 PPI.

**Table 4 materials-14-03153-t004:** The RMS values for different correlations.

Reference	RSM × 10^7^
Richardson et al. [39]	1.91
Improved Richardson et al.	1.35
Tadrist et al. [49]	1.82
Improved Tadrist et al.	1.27
Dukhan et al. [50]	2.14
Improved Dukhan et al.	1.59
Du Plessis et al. [51]	1.36
Calmidi [19]	1.37
Yang et al. [53]	4.58
Proposed Equation (20)	0.94
Proposed Equation (21)	1.22

## Data Availability

Data refer to the original literature.

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
