# Peer review of "(untitled)"

_materials, 2021, doi:10.3390/ma14123153_

Round 1
Reviewer 1 Report
Dear Authors,
Many thanks for your work. It was really interesting. However, it is very far from publication! I can not see any novel finding in this paper. Please find below comments on your paper:
- Figure 1 is wrong! Are you sure that you are reporting the correct Figure! I am not sure that one is open and other is closed!
- I have read the flow of your paper many parts are not connected well.
- The introduction is not specific and talks about every aspects!
- Only 1 figure belongs to your finding! Other figures are a comparison of others!
- The title must be changed to a kind of a review paper!
- I couldnt understand your finding is based on experiments or numerical analysis! It is a kind of RMS and curve fitting?
- How you have reached your model? You suddenly between the texts as some description for figure 6 you pointed out your model!
- I can not see your methodology!!!
Author Response
We would like to thank you for your effort to review the paper and provide the valuable comments. The comments have been considered carefully and the manuscript has been revised carefully according to reviewers’ comments. Below are answers addressing the specific comments:
Q1. Figure 1 is wrong! Are you sure that you are reporting the correct Figure! I am not sure that one is open and other is closed!
A: Sorry for the typo. It has been amended in the revised manuscript. Figure 1(a) is the closed-cell foam and citation sources is added. Figure 1(b) is the open-cell foam and it comes from our own results.
Q2. I have read the flow of your paper many parts are not connected well.
A: This paper mainly focuses on the pressure drop characteristics of flow through open-cell metal foam. The governing equations of fluid flow through open-cell metal foams was firstly reviewed and summarized. Secondly, the foam geometry structure that is the prerequisite for assessing the fluid flow transport through the open-cell metal foams was described. Thirdly, the pressure drop correlations for open-cell metal foams based on permeability and Forchheimer coefficient and friction factor were reviewed and summarized. Finally, the existing permeability and Forchheimer coefficient correlations were validated and improved against the experimental data. As a result, we develop and recommend three empirical correlations that have a higher forecasting precision than the previous correlations to predict the pressure drop through open-cell metal foam. The above four sections are all necessary to solve the pressure drop characteristics of flow through open-cell metal foam. As other reviewers’ opinions, the paper is well arranged with an understandable explanation. All sections of the paper are in correct format.
Q3. The introduction is not specific and talks about every aspects!
A: Thank you for your comment. The introduction is firstly described the skeleton characteristics of metal foam and its advantages. Then, we point out the questions from a considerable amount of work that a large discrepancy has been found between different studies and different theoretical models on the flow behavior such as the correlation between flow rate and pressure drop. In the following three sections, we will review and evaluate the previous works on the pressure drop characteristics of flow through open-cell metal foam. The introduction should be ok.
Q4. Only 1 figure belongs to your finding! Other figures are a comparison of others!
A: This paper aims to develop new pressure drop correlations for open-cell metal foam that bears an enhanced applicability and adaptability. Therefore, we put a lot of effort to evaluate the existing permeability and Forchheimer correlations against the experimental data collected in the open literatures. Therefore, it is understandable that only few figures are present to describe the improved correlations. The predictive ability by the improved correlations were added and evaluated in the revised manuscript.
Q5. The title must be changed to a kind of a review paper!
A: Thank you for your comment. Combined the opinions of other reviewers, the title was revised to Review and a theoretical approach on pressure drop correlations of flow through open-cell metal foam.
Q6. I couldnt understand your finding is based on experiments or numerical analysis! It is a kind of RMS and curve fitting?
A: The pressure drop correlations are all validated and improved against the experimental data collected in the open literatures that has been described on page 10-lines 288 through 290. The improved correlations are obtained based on the least square method. The performance of each correlations is assessed based on the RMS.
Q7. How you have reached your model? You suddenly between the texts as some description for figure 6 you pointed out your model!
A: Based on the comparison between the correlations and experimental measurements with the least square method, improvements of existing empirical correlations are proposed to enhance their estimation accuracy. It has been described on page 10-lines 288 through 290.
Q8. I can not see your methodology!
A: The governing equations of fluid flow through open-cell metal foam are obtained based on the volume averaging of conservation of mass for a fluid flowing in a porous medium. Therefore, the Darcy’s law and Forchheimer equation are the classic and universal equations.
Reviewer 2 Report
The manuscript is well arranged with an understandable explanation for new proposed correlations. In order for the manuscript to be accepted it is necessary revision. I have some specific remarks.
Best Regards
I suggest adding graphs representing a comparison between the results obtained by (old) correlations and the proposed correlations, not only with experimental results. In my opinion, this will add a new scientific value to the paper.
I suggest adding a table showing the new proposed correlations and comparing them to the old ones, and indicating their degree of accuracy. This table is a summary of the paper.
I suggest amended the main title to:
Assessment and Proposal of New correlations for Pressure Drop of Flow through Open-Cell Metal Foam: A Theoretical Approach
Author Response
Q1. I suggest adding graphs representing a comparison between the results obtained by (old) correlations and the proposed correlations, not only with experimental results. In my opinion, this will add a new scientific value to the paper.
A: We would like to thank you for your effort to review the paper and provide the valuable comments. The predicted permeability by these correlations were compared with the experimental data in the revised manuscript. It has been described on page 13 and 14.
Q1. I suggest adding a table showing the new proposed correlations and comparing them to the old ones, and indicating their degree of accuracy. This table is a summary of the paper.
A: The RMS values of these correlations were added and presented in the revised manuscript. It has been described on page 14.
Q1. I suggest amended the main title to: Assessment and Proposal of New correlations for Pressure Drop of Flow through Open-Cell Metal Foam: A Theoretical Approach
A: Thank you for your comment. Combined the opinions of other reviewers, the title was revised to Review and a theoretical approach on pressure drop correlations of flow through open-cell metal foam.
Reviewer 3 Report
The manuscript presents an interesting study of pressure drop in open-cell metal Foam. The authors proposed empirical equations based on Camidi model which sound useful for practical purposes. All sections of the paper are in correct format. As far as I know, Prof. Francesco Calomino published two articles about simulation of flow and pressure drop in a 'corrugated pipe' under 'free surface' and 'pressurized' flow conditions. I suppose those studies have some similarities with yours. So in your paper, you may utilize those studies to enhance accuracy and reliability of numerical models in prediction of pressure drop.
Author Response
Q1. The manuscript presents an interesting study of pressure drop in open-cell metal Foam. The authors proposed empirical equations based on Camidi model which sound useful for practical purposes. All sections of the paper are in correct format. As far as I know, Prof. Francesco Calomino published two articles about simulation of flow and pressure drop in a 'corrugated pipe' under 'free surface' and 'pressurized' flow conditions. I suppose those studies have some similarities with yours. So in your paper, you may utilize those studies to enhance accuracy and reliability of numerical models in prediction of pressure drop.
A: We would like to thank you for your effort to review the paper and provide the valuable comments. The two paper were added as references [23, 24] in the revised manuscript.
Reviewer 4 Report
This paper deals with the flow characteristics of fluids in open-cell metal foam. The fundamental governing equations of fluid flow through open-cell metal foams are summarized. The different pressure drop models for flow transport through open-cell foams are presented and discussed. In order to provide better prediction accuracy improvements are made on selected models using the least square method.
The manuscript represents a diligent and routine piece of work.
One major flaw was detected in the manuscript: Figure 1a and 1b show the same picture.
Author Response
Q1. One major flaw was detected in the manuscript: Figure 1a and 1b show the same picture.
A: We would like to thank you for your effort to review the paper and provide the valuable comments. Sorry for the typo. It has been amended in the revised manuscript.
Round 2
Reviewer 1 Report
Dear Authors,
Many thanks for your revised version. It doesnt change that much as I expected but according to the authors replies it seems that it is only a review paper so it must be categorize to review papers and must follow that structure!
Author Response
Q1. Many thanks for your revised version. It doesnt change that much as I expected but according to the authors replies it seems that it is only a review paper so it must be categorize to review papers and must follow that structure!
A: We would like to thank you for your effort to review the paper. This paper aims to develop new pressure drop correlations for open-cell metal foam that bears an enhanced applicability and adaptability. We put a lot of effort to review, evaluate and improve the existing permeability and Forchheimer correlations against the experimental data. Further, new correlations are developed and recommended for predicting the pressure drop of open-cell metal foams. Therefore, this manuscript is more than a review paper. I think the organization structure of this manuscript is reasonable.
Reviewer 2 Report
The suggested amendments have been included; in my opinion, the manuscript can be accepted.
All the best or authors
Author Response
Q1. The suggested amendments have been included; in my opinion, the manuscript can be accepted.
A: We would like to thank you for agreeing to accept this manuscript.